# Genome Sequencing of *Streptomyces olivaceus* SCSIO T05 and Activated Production of Lobophorin CR4 via Metabolic Engineering and Genome Mining

**DOI:** 10.3390/md17100593

**Published:** 2019-10-20

**Authors:** Chunyan Zhang, Wenjuan Ding, Xiangjing Qin, Jianhua Ju

**Affiliations:** 1CAS Key Laboratory of Tropical Marine Bio-resources and Ecology, Guangdong Key Laboratory of Marine Materia Medica, RNAM Center for Marine Microbiology, South China Sea Institute of Oceanology, Chinese Academy of Sciences, 164 West Xingang Road, Guangzhou 510301, China; zhchuny@foxmail.com (C.Z.); 13760785354@163.com (W.D.); xj2005qin@126.com (X.Q.); 2College of Oceanology, University of Chinese Academy of Sciences, 19 Yuquan Road, Beijing 100049, China

**Keywords:** genome sequencing, gene disruption, lobophorin, metabolic engineering, genome mining

## Abstract

Marine-sourced actinomycete genus *Streptomyces* continues to be an important source of new natural products. Here we report the complete genome sequence of deep-sea-derived *Streptomyces olivaceus* SCSIO T05, harboring 37 putative biosynthetic gene clusters (BGCs). A cryptic BGC for type I polyketides was activated by metabolic engineering methods, enabling the discovery of a known compound, lobophorin CR4 (**1**). Genome mining yielded a putative lobophorin BGC (*lbp*) that missed the functional FAD-dependent oxidoreductase to generate the d-kijanose, leading to the production of lobophorin CR4 without the attachment of d-kijanose to C17-OH. Using the gene-disruption method, we confirmed that the *lbp* BGC accounts for lobophorin biosynthesis. We conclude that metabolic engineering and genome mining provide an effective approach to activate cryptic BGCs.

## 1. Introduction

Microbially produced natural products (NPs) are an important reservoir of therapeutic and agricultural agents [1]. In the previous years, quantities of new bioactive NPs were isolated from marine-derived *Streptomyces* strains, suggesting marine-derived *Streptomyces* as a predominant source of new NPs [2]. In recent years, whole-genome sequencing programs have made it clear that microorganisms have greater biosynthetic potential but are mostly underexplored by virtue that most biosynthetic gene clusters (BGCs) in a single microbial genome are normally silent. Activation of these silent BGCs contributes to new NPs discoveries. Zhang and co-workers activated a cryptic polycyclic tetramate macrolactam (PTM) BGC in *Streptomyces pactum* SCSIO 02999 by promoter engineering and heterologous expression [3], and also promoted the expression of a silent PKS/NRPS hybrid BGC in the same *Streptomyces* strain by the alteration of several regulatory genes [4]. The production of nocardamine [5] and atratumycin [6] in *Streptomyces atratus* SCSIO ZH16 was turned on via metabolic engineering. These genome-based studies exemplify the benefits of genome mining and metabolic engineering used for activating cryptic BGCs and discovering new bioactive NPs. 

Lobophorins (Appendix A) belonging to a large class of spirotetronate antibiotics structurally feature a tetronate moiety *spiro*-linked with a cyclohexene ring, which is called pentacyclic aglycon or kijanolide [7,8,9,10,11,12,13,14,15,16,17]. Almost all of this class of compounds has a broad spectrum of antibacterial activities, as well as antitumor activity. Efforts to produce more spirotetronate antibiotics for drug discovery have thrived. Owing to the structural complexity of this family member, biosynthesis seems to be an effective way to afford the production of spirotetronate antibiotics, providing access to new analogues by pathway engineering and combinatorial biosynthetic approaches. In this paper, we report (i) the complete genome sequence of a deep-sea-derived *Streptomyces olivaceus* SCSIO T05, a talented strain capable of producing an array of putative NPs; (ii) activation of a cryptic lobophorin BGC (*lbp*) by mutagenetic methods and isolation of one known spirotetronate antibiotic lobophorin CR4 (**1**); and (iii) identification of the *lbp* BGC housed in *S. olivaceus* SCSIO T05 by gene-disruption experiment and bioinformatics analysis.

## 2. Results and Discussion

### 2.1. Genome Sequencing and Annotation of Streptomyces olivaceus SCSIO T05

Whole genome sequence is important when analyzing the potential production of secondary metabolites [5,18]. *S. olivaceus* SCSIO T05, a marine-derived strain, was previously reported to be isolated from the Indian Ocean deep-sea-derived sediment [19]. Its draft genome sequence was first gained by Illumina sequencing technology, but with several gap regions. In order to estimate the biosynthetic potential of *S. olivaceus* SCSIO T05, the complete genome was re-sequenced and acquired by the single-molecule real-time (SMRT) sequencing technology (PacBio). A total of 67156 filtered reads with high-quality data of 432570025 bp were generated, and then they were assembled into a linear contig by the hierarchical genome assembly process (HGAP) [20]. The complete genome revealed that 8458055 base pairs constitute a linear chromosome without a plasmid, with 72.51% of GC content (Figure 1 and Table 1). Totally, 7700 protein-coding genes were predicted, along with 18 rRNA and 65 tRNA. The genome sequence of *S. olivaceus* SCSIO T05 was deposited in GenBank (CP043317).

AntiSMASH analysis by using antiSMASH 5.0 [21] suggested 37 BGCs within the *S. olivaceus* SCSIO T05 genome (Figure 1 and Table 2). The 37 BGCs totally occupy 1.59 Mb, 18.76% of the complete genome. Most of the BGCs distribute in the two subtelomeric regions of the genome of some *Streptomyces* strains [18] and so do the BGCs in *S. olivaceus* SCSIO T05 genome. It is predicted that several BGCs are responsible for the production of polyketide- and nonribosome-peptide-derived secondary metabolites, including four PKS (Type I, Type II and Type III) and six NRPS, and six hybrid BGCs possess genes encoding more than one type of scaffold-synthesizing enzyme. Twenty-one BGCs are predicted to produce terpene, bacteriocin, lanthipeptide, or other categories. This analysis indicates that *S. olivaceus* SCSIO T05 is capable of producing an array of secondary metabolites, serving as a target strain for further metabolic engineering and genome mining.

### 2.2. Activation of a Cryptic Lobophorin BGC in the Genetically Engineered Mutant

In actuality, only a minority of potential chemicals are produced under standard laboratory culture conditions. Furthermore, the corresponding products are likely to be overlooked for multiple reasons, including low production rates, a large metabolic background, or improper culture conditions [22]. Fermented using modified-RA medium, the secondary metabolites produced by *S. olivaceus* SCSIO T05 were subsequently profiled using HPLC-DAD-UV. Multiple peaks were detected in the fermentation extract (Figure 2, trace i). We previously reported that five known NPs, rishirilides B (**2**) and C (**3**), lupinacidin A (**4**), galvaquinone B (**5**), and xiamycin A (**6**), were produced as major secondary metabolites from the wild-type strain [19,23]. In addition, an orphan dibenzoxazepinone biosynthetic pathway was mutagenically activated, leading to the production of new mycemycins [24], suggesting that *S. olivaceus* SCSIO T05 has a great potential for producing new NPs.

For exploring other secondary metabolites from the strain, *S. olivaceus* SCSIO T05/Δ*rsdK*_2_ (*S. olivaceus* SCSIO T05R) was constructed to abolish the production of the anthracenes [19]. The production of the second major secondary metabolites xiamycins was accumulated, along with a new peak around 26 min, distinct from the UV absorption characteristics of xiamycins (Figure 2, trace ii). For further background elimination of xiamycins, a “double-deletion” mutant *S. olivaceus* SCSIO T05/Δ*rsdK*_2_/Δ*xmcP* (*S. olivaceus* SCSIO T05RX) was constructed [23] in which the new peak (**1**) appeared to be the major product (Figure 2, trace iii). Accordingly, the *S. olivaceus* SCSIO T05RX mutant was fermented at a large scale, enabling the isolation and structure elucidation of this newly generated compound. It was identified as a known compound designated as lobophorin CR4 (Figure 3), by comparing HRESIMS, ^1^H, and ^13^C NMR data (Appendix A) to the reported data of an intermediate isolated from the *Streptomyces* sp. SCSIO 01127/Δ*lobG1* mutant [11]. It is reported that shifting metabolic flux of a wild-type strain by blocking the predominant product pathways may afford new secondary metabolites [5]. During our efforts to acquire new secondary metabolites by shifting the metabolic flux of marine actinomycetes [5,23,24], the production of nocardamine, olimycins, and mycemycins was turned on at the expense of major products by using gene knock-out methods. Similarly, the “double-deletion” mutant (*S. olivaceus* SCSIO T05RX) was constructed to abolish the production of two major secondary metabolites, anthracenes and xiamycins, from the wild-type strain [19,23]. With the engineered shifting of *S. olivaceus* metabolic flux, the newly produced lobophorin CR4 was activated.

### 2.3. Identification of a Putative Lobophorin (lbp) BGC via Genome Mining

The antiSMASH analysis of the complete genome of *S. olivaceus* SCSIO T05 revealed a 99.1 kb type I PKS BGC named as lobophorin BGC (*lbp*), showing highly similar traits to the reported *lob* BGCs from *Streptomyces* sp. FXJ7.023 [16] and *Streptomyces* sp. SCSIO 01127 [11]. The complete *lbp* contains 38 open reading frames (ORFs). The genetic organization of *lbp* is shown in Figure 4A, with genes color-coded on the basis of their proposed functions summarized in Table 3. The nucleotide sequences were deposited in GenBank (MN396889). The *lbp* BGC contains six inconsecutive genes *lbpA1*–*A6*, similar to *lobA1*–*A5* in *lob* from *S.* sp. SCSIO 01127. Differently, the LobA4 homologue is separated into two polyketide synthases (PKSs), LbpA4 and LbpA5, in *lbp*. The high similarity between the PKS modules in *lbp* and in *lob* enables us to propose that the assembly of the linear polyketide chain catalyzed by LbpA1–A6 utilizes six malonyl CoAs, six methylmalonyl-CoAs, and a 3-carbon glycerol unit (Figure 5) [11]. The *lbp* harbors four putative regulator genes (*lbpR1–R4*) (Figure 4 and Table 3) that are highly similar to *lobR1*, *lobR3*, *lobR4*, and *lobR5* in *lob*, respectively. These four regulators are assumed to be involved in the regulation network of lobophorin CR4 biosynthesis, which seems to be less complex than *lob* but more complex than *kij* [7] and *tca* [8]. In contrast, five regulator genes *lobR1–R5* are identified in *lob*; three regulator genes, *kijA8*, *kijC5*, and *kijD12,* are included in *kij*; *tcaR1* and *tcaR2* both encode regulators in *tca*. There is only one gene, *lbpU2* in *lbp*, with no apparent homologue in *lob* (Figure 4 and Table 3). The other genes included in *lbp* are putatively associated with the biosynthesis of kijanose and l-digitoxose units by virtue of high similarities to corresponding counterparts in *lob* (Figure 4 and Table 3). 

To demonstrate the validity of the putative *lbp* BGC, *lbpC4* coding for ketosynthase-III-like protein, which incorporates a 3-carbon glycerol unit into the biosynthetic precursor LOB aglycon [11], was disrupted by using PCR-targeting methods. As expected, the production of lobophorin CR4 was completely blocked in *S. olivaceus* SCSIO T05/Δ*rsdK*_2_/Δ*xmcP*/Δ*lbpC4* (*S. olivaceus* SCSIO T05RXL) (Figure 2, trace iv), demonstrating that the *lbp* BGC is indeed responsible for lobophorin biosynthesis. With high similarity to the *lob* BGC, the *lbp* BGC accounts for lobophorin CR4 without the attachment of kijanose to C17-OH, rather than lobophorins A and B in *lob*. Based on bioinformatics analysis, a series of enzymes are proposed to be involved in kijanose biosynthesis (Figure 5) [7]. Among them, the amino acid sequence of the putative FAD-dependent oxidoreductase LbpP2 is far shorter than its homologues LobP2 [11] and KijB3 [7]. KijB3 is proposed to oxidize the methyl group to a carboxylate group, essential for the generation of the kijanose moiety [7]. Multiple protein sequence alignments of LbpP2, LobP2, and KijB3 revealed that the conserved FAD binding domain is missing in LbpP2 (Appendix A). Thus, we speculate that LbpP2 is nonfunctional, failing to catalyze the carboxylation and hinder the generation of kijanose. 

Given the high similarity of LbpG3 and LobG3, we envision that LbpG3 has a similar function as LobG3, a glycosyltransferase from *S.* sp. SCSIO 01127, tandemly attaching the first two l-digitoxose at C-9 in lobophorins [11]. LbpG2 has 99% similarity to LobG2, another glycosyltransferase from the same strain, which was established to transfer the terminal l-digitoxose [11]. Both LbpG2 and LbpG3 are likely to be involved in the transfers of three sugar units, sugars A, B, and C, in lobophorin CR4 (Figure 5), consistent with the metabolite profile of Δ*lobG1* in *S.* sp. SCSIO 01127 [11].

## 3. Experimental Section

### 3.1. General Experimental Procedures

The plasmids and bacteria used are listed in Appendix A. *Streptomyces olivaceus* SCSIO T05 and its mutants were incubated on modified ISP-4 medium [25] with 3% sea salt and fermented in modified RA medium [19]. All cultures for *Streptomyces* were incubated at 28 °C. Luria-Bertani (LB) medium was used for *E. coli*, with appropriate antibiotics added at a final concentration of 100 µg/mL of ampicillin (Amp), 50 µg/mL of kanamycin (Kan), 50 µg/mL of apramycin (Apr), 25 µg/mL of chloroamphenicol (Cml), and 50 µg/mL of trimethoprim (TMP). 

A 1260 infinity system (Agilent, Santa Clara, CA, USA), which uses a Phenomenex Prodigy ODS (2) column (150 × 4.6 mm, 5 μm, USA), was used for HPLC-based analyses. Silica gel with the size of 100–200 mesh (Jiangpeng Silica gel development, Inc., Shandong, China) was used for column chromatography (CC). A Primaide 1110 solvent delivery module, which is equipped with a 1430 photodiode array detector (Hitachi, Tokyo, Japan) and uses a YMC-Pack ODS-A column (250 mm × 10 mm, 5 μm), was used for semi-preparative HPLC. A MaXis Q-TOF mass spectrometer (Bruker, Billerica, MA, USA) was used to acquire high-resolution mass spectral data. An MCP-500 polarimeter (Anton Paar, Graz, Austria) was used to record optical rotations. A Bruker Avance 500 was used to record NMR spectra. Carbon signals and the residual proton signals of DMSO-*d*_6_ were used for calibration (*δ*_C_ 39.52 and *δ*_H_ 2.50). 

### 3.2. Genome Sequencing and Bioinformatic Analysis

Whole genome scanning and annotation of *S. olivaceus* SCSIO T05 were acquired by the single-molecule real-time (SMRT) sequencing technology (PacBio) at Shanghai Majorbio Bio-Pharm Technology Co., Ltd (Shanghai, China). AntiSMASH (AntiSMASH 5.0, available at http://antismash.secondarymetabolites.org/) was used to analyze and assess the potential BGCs. FramePlot (FramePlot 4.0 beta, available at http://nocardia.nih.go.jp/fp4/) was used to analyze ORFs whose functions were predicted based on an online BLAST program (http://blast.ncbi.nlm.nih.gov/).

### 3.3. Construction of a “Triple-Deletion” Mutant Strain

Gene *lbpC4* from the *lbp* BGC was inactivated by the REDIRECT protocol [26]. All primers used in this study are listed in Appendix A. *LbpC4* was replaced by the apramycin resistance gene *oriT*/*aac(3)IV* fragment in the target cosmids 01-07D or 21-02E. The target mutant clones, *S. olivaceus* SCSIO T05RXL, were accomplished as previously described [19,23,24]. 

### 3.4. Fermentation and HPLC-based Analyses of S. olivaceus SCSIO T05 and Its Mutants

The *Streptomyces* used in this study were incubated in modified ISP-4 medium plates for 2–3 d. For fermentation, a portion of mycelium and spores was seeded into 50 mL of modified RA medium in a 250 mL flask and then shaken at 200 rpm and 28 °C for 8 d. The cultures were extracted with an equal volume of butanone. Organic phases were then dissolved in CH_3_OH (1 mL) after having been evaporated to dryness, and 40 µL of each relevant sample was injected for HPLC-based analysis. The UV detection was at 254 nm. Solvent A is composed of 85% ddH_2_O and 15% CH_3_CN, supplemented with 0.1% HOAc. Solvent B is composed of 85% CH_3_CN and 15% ddH_2_O, supplemented with 0.1% HOAc. Samples were analyzed via the following method: a linear gradient from 0% to 80% solvent B in 20 min, and then, from 80% to 100% solvent B for 1.5 min, finally eluted with 100% solvent B in 6.5 min. The flow rate was 1.0 mL/min.

### 3.5. Production, Isolation, and Structure Elucidation of Lobophorin CR4

The mycelium of *S. olivaceus* SCSIO T05RX were inoculated into 50 mL of modified-RA medium and then shaken at 200 rpm and 28 °C for 2 d, to gain the seed cultures. After that, the seed cultures were transferred into 150 mL of modified-RA medium and shaken at 200 rpm and 28 °C for 8 d. After the large-scale fermentation was accomplished, a total of 12 L of the growth culture was centrifuged at 4000 g for 10 min to separate the supernatant and mycelium and further extracted by butanone and acetone, respectively. The two organic phases were concentrated (via solvent removal under vacuum), and the residues were combined. The combined sample was subjected to normal phase silica gel CC eluted with CHCl_3_-CH_3_OH (100:0, 98:2, 96:4, 94:6, 92:8, 90:10, 85:15, 80:20, 70:30, 50:50, v/v, each solvent combination in 250 mL volume) to give ten fractions (AFr.1–AFr.10). Fractions A1-A3 were purified to afford the accumulation of compound **1** (98 mg), by preparative HPLC, eluting with 90% solvent B (A: H_2_O; B: CH_3_CN) over the course of 30 min. The flowrate was 2.5 mL/min and the UV detection was at 254 nm. The purified compound was subjected to MS, ^1^H, and ^13^C NMR spectra measurements and elucidated as a known intermediate **3** during lobophorins A and B biosynthesis [11], and we named it lobophorin CR4 (**1**). 

## 4. Conclusions

In this study, we acquired the complete genome sequence of *S. olivaceus* SCSIO T05. The biosynthetically talented strain harbors 37 putative BGCs analyzed by antiSMASH. To explore the biosynthetic potential of this strain, metabolic engineering and genome mining were performed. The major anthracenes and indolosesquiterpenes biosynthetic pathways were blocked, and an orphan spirotetronate antibiotics BGC (*lbp*) was activated in *S. olivaceus* SCSIO T05, leading to the isolation and identification of one known compound, lobophorin CR4. We have identified the *lbp* BGC accounting for lobophorin biosynthesis by gene-disruption experiments and bioinformatics analysis. The production of lobophorin CR4 without the attachment of d-kijanose to C17-OH was on account that the nonfunctional FAD-dependent oxidoreductase LbpP2 failed to generate d-kijanose. This work highlights that metabolic engineering and genome mining are the effective ways to turn on putative orphan or silent BGCs to acquire new NPs for drugs discovery.

## Figures and Tables

**Figure 1 marinedrugs-17-00593-f001:**
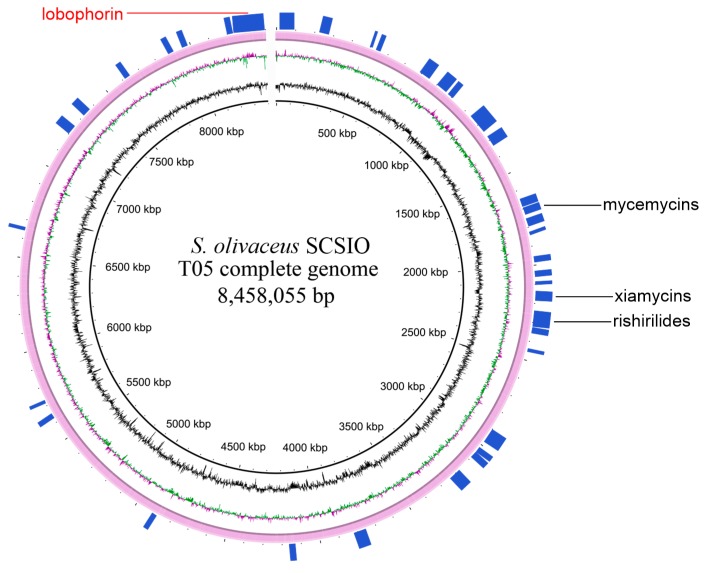
The complete genome of *S. olivaceus* SCSIO T05. The three circles (inner to outer) represent forward GC content, GC skew, and the distribution of putative biosynthetic gene clusters (BGCs) (represented by the bars) generated by antiSMASH 5.0. Clusters 18, 17, and 11 were described as rishirilides, xiamycins, and mycemycins BGCs, respectively. The putative lobophorin BGC with red color was referred to as cluster 37.

**Figure 2 marinedrugs-17-00593-f002:**
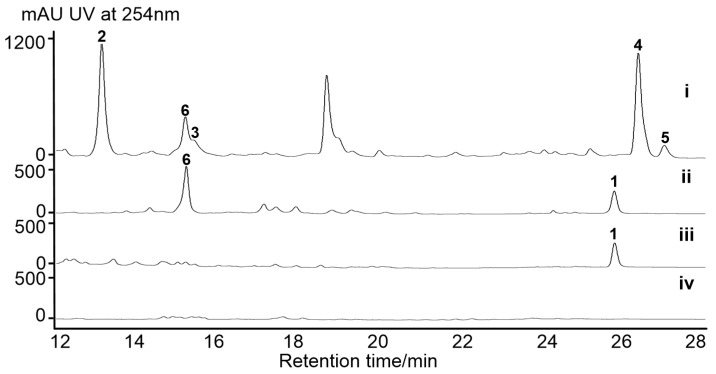
HPLC-based analyses of fermentation broths: (i) *S. olivaceus* SCSIO T05; (ii) *S. olivaceus* SCSIO T05R; (iii) *S. olivaceus* SCSIO T05RX; and (iv) *S. olivaceus* SCSIO T05RXL. Compound **1** is lobophorin CR4. Compounds **2**–**6** were previously identified as rishirilide B, rishirilide C, lupinacidin A, galvaquinone B, and xiamycin A, respectively.

**Figure 3 marinedrugs-17-00593-f003:**
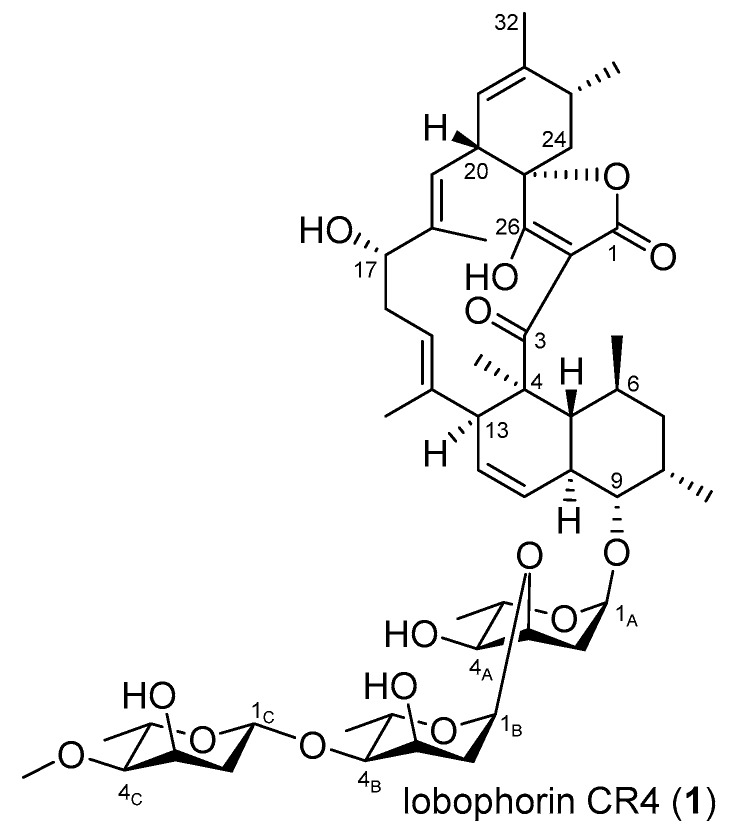
Structure of the isolated lobophorin CR4.

**Figure 4 marinedrugs-17-00593-f004:**
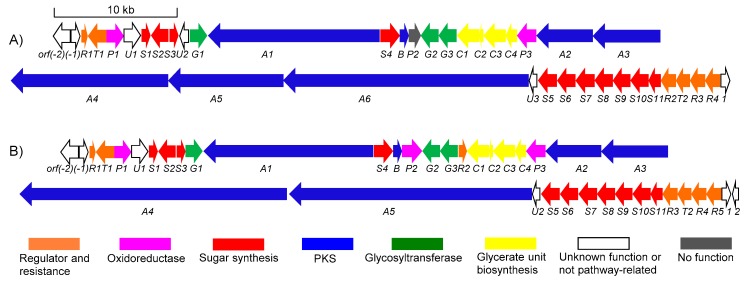
Genetic organizations: (**A**) the *lbp* BGC from *S. olivaceus* SCSIO T05; (**B**) the *lob* BGC from *S.* sp. SCSIO 01127.

**Figure 5 marinedrugs-17-00593-f005:**
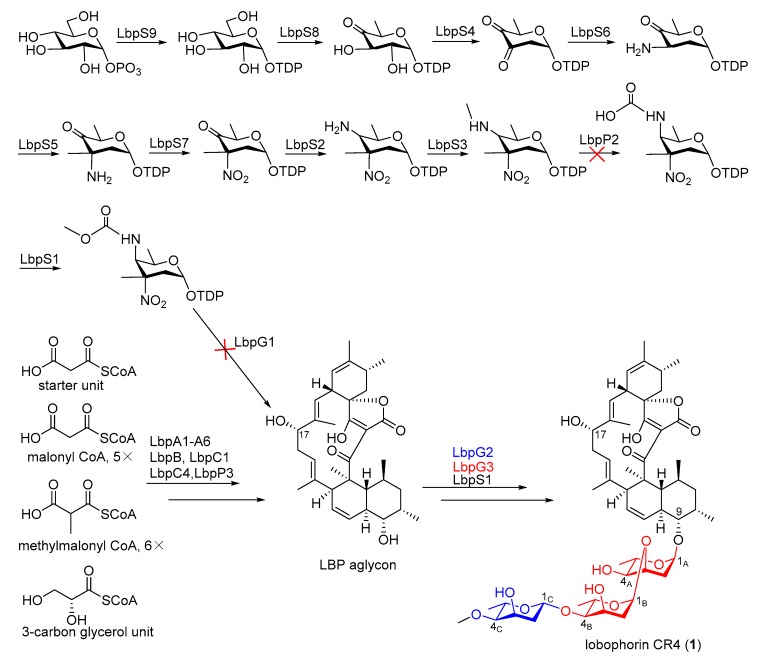
Proposed biosynthetic pathway of lobophorin CR4.

**Table 1 marinedrugs-17-00593-t001:** Genome features of *S. olivaceus* SCSIO T05.

Feature	Value
Genome size (bp)	8,458,055
Average GC content (%)	72.51
Protein-coding genes	7700
Total size of Protein-coding genes (bp)	7,543,173
rRNAs number	18
tRNAs number	65

**Table 2 marinedrugs-17-00593-t002:** AntiSMASH-predicted BGCs for *S. olivaceus* SCSIO T05.

BGC	Position		Type (Product)
	From	To	
Cluster 1	2725	89768	Type I Polyketide synthase (T1 PKS)
Cluster 2	234616	284137	Non-ribosomal peptide synthetase (NRPS) cluster
Cluster 3	504553	512728	Bacteriocin or other unspecified ribosomally synthesized and post-translationally modified peptide product (RiPP) cluster (Bacteriocin)
Cluster 4	525945	544617	Terpene
Cluster 5	793277	855894	NRPS
Cluster 6	901333	979368	T1 PKS
Cluster 7	980891	1005613	Lanthipeptide cluster (Lanthipeptide)
Cluster 8	1135886	1240760	Other types of PKS cluster (Otherks)-NRPS
Cluster 9	1275164	1347740	NRPS-Terpene
Cluster 10	1651711	1694648	NRPS-Nucleoside cluster (Nucleoside)
Cluster 11	1695020	1734380	Otherks
Cluster 12	1751698	1796277	NRPS
Cluster 13	1840051	1851963	Siderophore cluster (Siderophore)
Cluster 14	1967451	1990613	Lanthipeptide
Cluster 15	2037772	2059400	Terpene
Cluster 16	2090680	2102023	Bacteriocin
Cluster 17	2138860	2187226	T1PKS-NRPS
Cluster 18	2230691	2317060	NRPS-Type II PKS (T2 PKS)-Otherks
Cluster 19	2330735	2352337	Lanthipeptide
Cluster 20	2443907	2456009	Siderophore
Cluster 21	2905748	2978302	T2 PKS
Cluster 22	3029068	3048760	Terpene
Cluster 23	3049806	3075321	Beta-lactone containing protease inhibitor (Betalactone)
Cluster 24	3182776	3235915	NRPS
Cluster 25	3764472	3822515	NRPS
Cluster 26	4131410	4159582	Lanthipeptide
Cluster 27	4881296	4901736	Phenazine cluster (Phenazine)
Cluster 28	5633979	5656500	Lasso peptide cluster (Lassopeptide)
Cluster 29	5716930	5727556	Melanin cluster (Melanin)
Cluster 30	6667385	6677783	Ectoine cluster (Ectoine)
Cluster 31	7200930	7253804	NRPS
Cluster 32	7328818	7368924	Type III PKS (T3 PKS)
Cluster 33	7614814	7636052	Aminoglycoside/aminocyclitol cluster (Amglyccycl)
Cluster 34	7882883	7906528	Terpene
Cluster 35	7959695	7980831	Indole cluster (Indole)
Cluster 36	8200560	8221618	Terpene
Cluster 37	8239655	8455702	T1pks-Nrps-T3 PKS-Oligosaccharide cluster (Oligosaccharide)-Other

**Table 3 marinedrugs-17-00593-t003:** Deduced function of open reading frames (ORFs) in the *lbp* BGC.

ORF	Size ^a^	Proposed Function	ID/SI ^b^	Protein Homologue and Origin
*orf(-2)*	374	macrolide glycosyltransferase	100/100	Orf(-2) (AGI99472.1); *Streptomyces* sp. SCSIO 01127
*orf(-1)*	260	FkbM family methyltransferase	100/100	Orf(-1) (AGI99473.1); *Streptomyces* sp. SCSIO 01127
*lbpR1*	195	TetR type regulatory protein	100/100	lobR1 (AGI99474.1); *Streptomyces* sp. SCSIO 01127
*lbpT1*	497	efflux permease	100/100	lobT1 (AGI99475.1); *Streptomyces* sp. SCSIO 01127
*lbpP1*	392	p450 monooxygenase	100/100	lobP1 (AGI99476.1); *Streptomyces* sp. SCSIO 01127
*lbpU1*	326	aldo/keto reductase	100/100	lobU1 (AGI99477.1); *Streptomyces* sp. SCSIO 01127
*lbpS1*	271	sugar-O-methyltransferase	99/100	lobS1 (AGI99478.1); *Streptomyces* sp. SCSIO 01127
*lbpS2*	384	sugar 4-aminotransferase	100/100	lobS2 (AGI99479.1); *Streptomyces* sp. SCSIO 01127
*lbpS3*	201	SAM-dependent methyltransferase	97/98	lobS3 (AGI99480.1); *Streptomyces* sp. SCSIO 01127
*lbpU2*	197	hypothetical protein	100/100	hypothetical protein (KMB22099.1); Klebsiella pneumoniae
*lbpG1*	391	glycosyltransferase	100/100	lobG1 (AGI99481.1); *Streptomyces* sp. SCSIO 01127
*lbpA1*	3936	PKS (KS-AT-DH-ER-KR-ACP-KS-AT-DH-KR-ACP)	100/100	lobA1 (AGI99482.1); *Streptomyces* sp. SCSIO 01127
*lbpS4*	483	sugar 2,3-dehydratase	100/100	lobS4 (AGI99483.1); *Streptomyces* sp. SCSIO 01127
*lbpB*	253	thioesterase	100/100	lobB (AGI99484.1); *Streptomyces* sp. SCSIO 01127
*lbpP2*	313	FAD-dependent oxidoreductase	100/100	part of lobP2 (AGI99485.1); *Streptomyces* sp. SCSIO 01127
*lbpG2*	416	glycosyltransferase	99/100	lobG2 (AGI99486.1); *Streptomyces* sp. SCSIO 01127
*lbpG3*	476	glycosyltransferase	99/100	lobG3 (AGI99487.1); *Streptomyces* sp. SCSIO 01127
*lbpC1*	680	hydrolase superfamily dihydrolipo-amide acyltransferase-like protein	99/99	lobC1 (AGI99489.1); *Streptomyces* sp. SCSIO 01127
*lbpC2*	75	ACP	99/100	lobC2 (AGI99490.1); *Streptomyces* sp. SCSIO 01127
*lbpC3*	621	FkbH-like protein	99/100	lobC3 (AGI99491.1); *Streptomyces* sp. SCSIO 01127
*lbpC4*	342	ketoacyl acyl carrier protein synthase III	100/100	lobC4 (AGI99492.1); *Streptomyces* sp. SCSIO 01127
*lbpP3*	492	FAD-dependent oxidoreductase	100/100	lobP3 (AGI99493.1); *Streptomyces* sp. SCSIO 01127
*lbpA2*	1573	PKS (KS-AT-KR-ACP)	99/100	lobA2 (AGI99494.1); *Streptomyces* sp. SCSIO 01127
*lbpA3*	1798	PKS (KS-AT-DH-KR-ACP)	99/99	lobA3 (AGI99495.1); *Streptomyces* sp. SCSIO 01127
*lbpA4*	4376	PKS (KR-ACP-KS-AT-DH-KR-ACP-KS-AT-DH-KR-ACP)	100/100	part of lobA4 (AGI99496.1); *Streptomyces* sp. SCSIO 01127
*lbpA5*	2881	PKS (KS-AT-DH-KR-ACP-KS-AT-DH)	99/98	part of lobA4 (AGI99496.1); *Streptomyces* sp. SCSIO 01127
*lbpA6*	6362	PKS (KS-AT-ACP-KS-AT-DH-KR-ACP-KS-AT-DH-KR-ACP-KS-AT-DH-KR-ACP)	99/99	lobA5 (AGI99497.1); *Streptomyces* sp. SCSIO 01127
*lbpU3*	151	unknown	100/100	lobU2 (AGI99498.1); *Streptomyces* sp. SCSIO 01127
*lbpS5*	414	sugar 3-C-methyl transferase	100/100	lobS5 (AGI99499.1); *Streptomyces* sp. SCSIO 01127
*lbpS6*	373	sugar 3-aminotransferase	100/100	lobS6 (AGI99500.1); *Streptomyces* sp. SCSIO 01127
*lbpS7*	439	acyl-CoA dehydrogenase	100/100	lobS7 (AGI99501.1); *Streptomyces* sp. SCSIO 01127
*lbpS8*	341	sugar 4,6-dehydratase	100/100	lobS8 (AGI99502.1); *Streptomyces* sp. SCSIO 01127
*lbpS9*	298	sugar nucleotidyltransferase	99/100	lobS9 (AGI99503.1); *Streptomyces* sp. SCSIO 01127
*lbpS10*	332	sugar 3-ketoreductase	100/100	lobS10 (AGI99504.1); *Streptomyces* sp. SCSIO 01127
*lbpS11*	202	sugar 5-epimerase	99/100	lobS11 (AGI99505.1); *Streptomyces* sp. SCSIO 01127
*lbpR2*	274	TetR type regulatory protein	99/100	lobR3 (AGI99506.1); *Streptomyces* sp. SCSIO 01127
*lbpT2*	211	forkhead-associated protein	99/100	lobT2 (AGI99507.1); *Streptomyces* sp. SCSIO 01127
*lbpR3*	298	putative regulatory protein	99/100	lobR4 (AGI99508.1); *Streptomyces* sp. SCSIO 01127
*lbpR4*	309	LysR family transcriptional regulator	99/100	lobR5 (AGI99509.1); *Streptomyces* sp. SCSIO 01127
*orf1*	183	acetyltransferase	100/100	Orf1 (AGI99510.1); *Streptomyces* sp. SCSIO 01127

^a^ Amino acids. ^b^ Identity/similarity.

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
