# Peer review of "Genome Sequencing of Streptomyces olivaceus SCSIO T05 and Activated Production of Lobophorin CR4 via Metabolic Engineering and Genome Mining"

_marinedrugs, 2019, doi:10.3390/md17100593_

Round 1

Reviewer 1 Report

In the manuscript entitled, "Genome Sequencing of Streptomyces olivaceus SCSIO T05 and Activated Production of Lobophorin CR4 via Metabolic Engineering and Genome Mining" submitted by Zhang et al. the authors present the biosynthetic potential of a newly sequenced marine Streptomyces. Following a series of biosynthetic pathway knock-outs, the authors observe the production of a known metabolite, lobophorin. Considering the sequenced Streptomyces was observed to also possess the previously characterized lobophorin biosynthetic gene cluster, this result is not surprising. Overall, the manuscript describes a couple of knock-out mutant strains that appear to start producing Lobophorin. While the results are significant and likely of interest to the Marine Drugs readership, as written the manuscript requires more discussion about the results and minor revisions before acceptance. Below are some questions and concerns for the reviewers.

While it is clear that the KO mutants do produce detectable quantities of lobophorin, the blasé explanation for this result is poorly communicated. As this is the most significant result, the authors should provide more detail concerning their results. Do the authors consider this to be a mere consequence of metabolic flux and increased pools of CoA esters? Or might the detectable levels of lobophorin be due to diminished noise in the LC-MS analysis? Do the authors have references for either case? Is it typical to see increased/activated production upon knocking out active BGCs? The use of the phrase "metabolic engineering" suggests that it was the intention of the authors to facilitate this result. If this is the case, there should be considerable explanation/discussion about the activated production of lobophorin. 

Author Response

Comments from Reviewer 1:

Comments and Suggestions for Authors:

In the manuscript entitled, "Genome Sequencing of Streptomyces olivaceus SCSIO T05 and Activated Production of Lobophorin CR4 via Metabolic Engineering and Genome Mining" submitted by Zhang et al. the authors present the biosynthetic potential of a newly sequenced marine Streptomyces. Following a series of biosynthetic pathway knock-outs, the authors observe the production of a known metabolite, lobophorin. Considering the sequenced Streptomyces was observed to also possess the previously characterized lobophorin biosynthetic gene cluster, this result is not surprising. Overall, the manuscript describes a couple of knock-out mutant strains that appear to start producing Lobophorin. While the results are significant and likely of interest to the Marine Drugs readership, as written the manuscript requires more discussion about the results and minor revisions before acceptance. Below are some questions and concerns for the reviewers.

1) While it is clear that the KO mutants do produce detectable quantities of lobophorin, the blasé explanation for this result is poorly communicated. As this is the most significant result, the authors should provide more detail concerning their results. Do the authors consider this to be a mere consequence of metabolic flux and increased pools of CoA esters? Or might the detectable levels of lobophorin be due to diminished noise in the LC-MS analysis? Do the authors have references for either case?

Response: We thank the reviewer for this query and appreciate this excellent suggestion. We envision that the production of lobophorin CR4 was activated owing to the engineered shifting of S. olivaceus metabolic flux. More blasé explanation for this result was added in the main text, Page 5.

To acquire new secondary metabolites by shifting metabolic flux of marine actinomycetes, the production of nocardamine (Li, Y.; Zhang, C.; Liu, C.; Ju, J.; Ma, J. Front. Microbiol. 2018, 9, 1269.), olimycins (Sun, C.; Zhang, C.; Qin, X.; Wei, X.; Liu, Q.; Li, Q.; Ju, J. Tetrahedron. 2018, 74, 199–203.), and mycemycins (Zhang, C.; Yang, Z.; Qin, X.; Ma, J.; Sun, C.; Huang, H.; Li, Q.; Ju, J. Org. Lett. 2018, 20, 7633–7636.), was activated at the expense of major products by using gene knock-out methods. We have cited these three references in the main text, Page 5.

2) Is it typical to see increased/activated production upon knocking out active BGCs?

Response: We thank the reviewer for this query. Knocking out active BGCs is one way to activate the production of other products that can be used, as what we have noted in response to comment 1 of reviewer 1. Sometimes further performances, such as changing culture conditions, overexpression of pathway-specific positive regulatory genes or promotor constructions, are called for activating the production or enhancing the yield.

3) The use of the phrase "metabolic engineering" suggests that it was the intention of the authors to facilitate this result. If this is the case, there should be considerable explanation/discussion about the activated production of lobophorin. 

Response: We appreciate this excellent suggestion. We have noted it in response to comment 1 of reviewer 1. These words were added in the main text, Page 5.

Reviewer 2 Report

I find that the paper needs some revisions, and here some suggestions to improve the text.

The introduction needs revision. the first paragraph contains a very long sentence, very difficult to understand.It is also not clear why do they talk about synthetic efforts and biosynthesis, they are not synonymous.The expression "new comers" needs to be corrected

Results and discussion are presented together, the text is divided in 3 different paragraphs, but grafically it is difficult to distinguish titles form the text. Sometimes the reasoning behind some experiments is not clear, for example about the deletion strain constructed. I would also suggest to change the name of the compound from "1" to something less confusing.Images and tables needs to have longer legends in order to understand the meaning of what is shown and how the researchers got there. So overall, the text needs to be enriched.

Materials and methods are placed after results and discussion but before Conclusions. Is that standard for the journal? Usually they are placed of before the results, or after the conclusions.

Author Response

Comments from Review 2:

Comments and Suggestions for Authors:

I find that the paper needs some revisions, and here some suggestions to improve the text.

1) The introduction needs revision. The first paragraph contains a very long sentence, very difficult to understand.

Response: We have revised this long sentence in the first paragraph to read as follows: “Activation of these silent BGCs contributes to new NPs discovery. Zhang and co-workers activated a cryptic polycyclic tetramate macrolactam (PTM) BGC in Streptomyces pactum SCSIO 02999 by promoter engineering and heterologous expression [3], and also promoted expression of a silent PKS/NRPS hybrid BGC in the same Streptomyces strain by alteration of several regulatory genes [4]. The production of nocardamine [5] and atratumycin [6] in Streptomyces atratus SCSIO ZH16 were turned on via metabolic engineering. These genome-based studies exemplify the benefits of genome mining and metabolic engineering used for activating the cryptic BGCs and discovering new bioactive NPs.

2) It is also not clear why do they talk about synthetic efforts and biosynthesis, they are not synonymous. The expression "new comers" needs to be corrected.

Response: We thank the reviewer for this query. We have revised this sentence in the second paragraph to read as follows: “Efforts to produce more spirotetronate antibiotics for drug discovery have thrived”. And we have revised “new comers” to “new analogues”.

3) Results and discussion are presented together, the text is divided in 3 different paragraphs, but grafically it is difficult to distinguish titles form the text.

Response: We have separated the subtitles from the text to read as follows: “2.1. Genome Sequencing and Annotation of Streptomyces olivaceus SCSIO T05”, “2.2. Activation of a Cryptic Lobophorin BGC in the Genetic Engineered Mutant”, and “2.3. Identification of a Putative Lobophorin (lbp) BGC by Genome Mining”.

4) Sometimes the reasoning behind some experiments is not clear, for example about the deletion strain constructed.

Response: We appreciate this excellent suggestion. The “double-deletion” mutant (S. olivaceus SCSIO T05RX) was constructed to abolish the production of two major secondary metabolites, anthracenes and xiamycins from the wild-type strain, S. olivaceus SCSIO T05. These words were added in the main text, Page 5.

5) I would also suggest to change the name of the compound from "1" to something less confusing.

Response: We appreciate this excellent suggestion. We have revised “compound 1” to “lobophorin CR4” throughout the manuscript.

6) Images and tables needs to have longer legends in order to understand the meaning of what is shown and how the researchers got there. So overall, the text needs to be enriched.

Response: We appreciate this excellent suggestion. We have added notes in the figure legend as suggested.

7) Materials and methods are placed after results and discussion but before Conclusions. Is that standard for the journal? Usually they are placed of before the results, or after the conclusions.

Response: We followed the Marine Drugs Instructions for Authors to place the research manuscript sections. Materials and methods are placed after results and discussion, and before Conclusions.

Reviewer 3 Report

The paper is well explained and the authors have used leading methodologies. However, I think the introduction could be improved. As an example I wouldn't say "marine-sourced actinomycete genus Streptomyces continue to be the PREDOMINANT source of new natural products" which is actually repeated twice in the paper (Abstract and introduction), due to the wide biodiversity in which they are founded (fungi, plants, soil, ants, algae, etc). On the other hand, it would be appreciated to name which specific instruments have been used for the MS, and the high resolution ionization MS in methods.

I think here has been proved that, by metabolic engineering and genome mining, silent BGCs can be turn on, unfortunately in this case led us to a known compound. Future work in the triple mutant as an OSMAC approach with different carbon sources could be successful.

Author Response

Comments from Review 3:

Comments and Suggestions for Authors:

The paper is well explained and the authors have used leading methodologies. However, I think the introduction could be improved.

1) As an example I wouldn't say "marine-sourced actinomycete genus Streptomyces continue to be the PREDOMINANT source of new natural products" which is actually repeated twice in the paper (Abstract and introduction), due to the wide biodiversity in which they are founded (fungi, plants, soil, ants, algae, etc).

Response: We appreciate this excellent suggestion. We have revised the sentence in the introduction to read as follows: “quantities of new bioactive NPs have been isolated from marine-derived Streptomyces strains, suggesting marine-derived Streptomyces as a predominant source of new NPs.”

2) On the other hand, it would be appreciated to name which specific instruments have been used for the MS, and the high resolution ionization MS in methods.

Response: We have mentioned that High-resolution mass spectral data were obtained on a MaXis Q-TOF mass spectrometer in General Experimental Procedures, Experimental Section.

3) I think here has been proved that, by metabolic engineering and genome mining, silent BGCs can be turn on, unfortunately in this case led us to a known compound. Future work in the triple mutant as an OSMAC approach with different carbon sources could be successful.

Response: We appreciate this excellent suggestion. We will try further performances, such as changing culture conditions, overexpression of pathway-specific positive regulatory genes and promotor constructions, to trigger more secondary metabolites biosynthesis in this triple-deletion mutant.

Reviewer 4 Report

Several typos and grammatically incorrect or not quite clear sentences need to be addresses. They have been highlighted in the attached file.

Fig. 1 is not mentioned in the text.

p.2 SM acronym appears without introducing what it stands for.

Fig. 2. Why are three BGCs highlighted and not the rest? Why is lobophorin a different colour? The legend is not explanatory of what can be seen in the figure. This figure and legend need improvement to be clear.

The identification and characterization of the lbp BGC section lacks experimental evidence. There is no experimental evidence presented to support the authors’ proposed polyketide assembly line, or any experimental evidence to support that the predicted regulators are in fact acting as regulators of the BGC. The authors use statements such as encode when they mean they are predicted to, without any experimental evidence. In this section, only one gene has been disrupted experimentally and therefore the authors have not characterised the BGC but have established that the product of that particular gene is involved in the synthesis of the compound. These are completely different things.

In order to characterise the BGC the authors need to identify the minimum number of genes required to produce the compound, they should do this by deletion of all the genes within the cluster, to prove that in fact all of the genes in the cluster are required for the synthesis of the compound. Otherwise, characterisation of the cluster cannot be claimed, as all the information presented is dependent on a bioinformatic prediction.

Methods section is too vague.

Table S1. Authors claim that most strains and plasmids where generated in this study but this is not true. It seems that only plasmids p01-07D p21-02E have been generated in this study. All other plasmids have been generated by different research groups and have been widely used by the community in the last 15 years. All other E. coli strains have been generated by different research groups and have been widely used by the community for more than 15 years. The S. olivaceus strains mentioned except S. olivaceus SCSIO T05RXL seem to have been generated by this research group prior to this study.

This study does not seem to be novel enough and lacks the experimental data to prove that the predicted BGC is responsible for the synthesis of lobophorin.

Author Response

Comments from Review 4:

Comments and Suggestions for Authors:

Several typos and grammatically incorrect or not quite clear sentences need to be addresses. They have been highlighted in the attached file.

Response: We appreciate this excellent suggestion. We have revised the manuscript as suggested in the attached file.

1) Fig. 1 is not mentioned in the text.

Response: We have revised “Fig. 1” to “Fig. 3” and mentioned it in the manuscript, Page 5.

2) p.2 SM acronym appears without introducing what it stands for.

Response: We have revised “SMs” acronym to “secondary metabolites” throughout the manuscript.

3) Fig. 2. Why are three BGCs highlighted and not the rest? Why is lobophorin a different colour? The legend is not explanatory of what can be seen in the figure. This figure and legend need improvement to be clear.

Response: We thank the reviewer for these queries. We have added a note in the figure legend as suggested in Page 3.

4) The identification and characterization of the lbp BGC section lacks experimental evidence. There is no experimental evidence presented to support the authors’ proposed polyketide assembly line, or any experimental evidence to support that the predicted regulators are in fact acting as regulators of the BGC. The authors use statements such as encode when they mean they are predicted to, without any experimental evidence. In this section, only one gene has been disrupted experimentally and therefore the authors have not characterised the BGC but have established that the product of that particular gene is involved in the synthesis of the compound. These are completely different things. In order to characterise the BGC the authors need to identify the minimum number of genes required to produce the compound, they should do this by deletion of all the genes within the cluster, to prove that in fact all of the genes in the cluster are required for the synthesis of the compound. Otherwise, characterisation of the cluster cannot be claimed, as all the information presented is dependent on a bioinformatic prediction.

Response: We appreciate this excellent suggestion. We have revised the description as suggested in Section “2.3. Identification of a Putative Lobophorin (lbp) BGC by Genome Mining”, pages 5, 6, and 7. We identified the putative lobophorin (lbp) BGC by genome mining, which coincides with the title of this paper. The involvement of the lbp gene cluster in the biosynthesis of lobophorin CR4 was further demonstrated by inactivating the ketoacyl acyl carrier protein synthase III gene (lbpC4) to generate a lobophorin CR4 non-producing mutant (S. olivaceus SCSIO T05RXL). The high similarity between the lbp and lob biosynthetic gene clusters enabled us to propose a biosynthetic pathway of lobophorin CR4.

5) Methods section is too vague. Table S1. Authors claim that most strains and plasmids where generated in this study but this is not true. It seems that only plasmids p01-07D p21-02E have been generated in this study. All other plasmids have been generated by different research groups and have been widely used by the community in the last 15 years. All other E. coli strains have been generated by different research groups and have been widely used by the community for more than 15 years. The S. olivaceus strains mentioned except S. olivaceus SCSIO T05RXL seem to have been generated by this research group prior to this study.

Response: We have revised the information of strains and plasmids in Supporting Information.

6) This study does not seem to be novel enough and lacks the experimental data to prove that the predicted BGC is responsible for the synthesis of lobophorin.

Response: We thank the reviewer. Lobophorins are an important group of spirotetronate antibiotics with a broad spectrum of antibacterial activities, as well as antitumor activity. In this manuscript, we focus on the novel findings that justify publication in Marine Drugs: i) the biosynthetic potential of marine-derived Streptomyces olivaceus SCSIO T05 by antiSMASH analysis of the complete genome sequence; ii) the activation of a cryptic lobophorin gene cluster by genetic engineering methodology, enabling the discovery of lobophorin CR4; iii) genome mining of lbp gene cluster accounting for lobophorin biosynthesis by gene-disruption experiments and bioinformatics analysis. This work highlights metabolic engineering and genome mining as effective ways to activate putative orphan or silent gene clusters to acquire new natural products for drugs discovery.

Sequence analysis indicated that the predicted lbp gene cluster from Streptomyces olivaceus SCSIO T05 and the lob gene cluster from Streptomyces sp. SCSIO 01127 exhibited striking similarities and shared a vast majority of homologues, suggesting that the lbp gene cluster might account for lobophorin CR4 biosynthesis in Streptomyces olivaceus SCSIO T05. The involvement of the lbp gene cluster in the biosynthesis of lobophorin CR4 was further demonstrated by inactivating the ketoacyl acyl carrier protein synthase III gene (lbpC4) to generate a lobophorin CR4 non-producing mutant (S. olivaceus SCSIO T05RXL).

This manuscript is a resubmission of an earlier submission. The following is a list of the peer review reports and author responses from that submission.